# Leverage of Local State-Owned Enterprises, Implicit Contingent Liabilities of Government and Economic Growth

**Yixuan Duan** [1,*]**, Min Guo** [2] **and Yixuan Huang** [3]

1   School of Economics and Management, China University of Petroleum, Beijing 102249, China
2   School of Banking & Finance, University of International Business and Economics, Beijing 100029, China; guomin@uibe.edu.cn
3   Research Center for Finance, Chinese Academy of Fiscal Sciences, Beijing 100142, China; huangyixuan@chineseafs.org
*   Correspondence: 13120025877@163.com

**Abstract:** Local state-owned enterprises (SOEs) working together with local governments can promote economic growth. However, an increase in the implicit contingent liabilities of local governments due to implicit guarantees given to SOEs has a negative effect on economic growth. The classical socialist theories and the economic stability in each financial crisis of China show that the macroeconomic efficiency of SOEs is more important than the microeconomic efficiency, and microeconomic efficiency in neoclassical economic theory cannot reflect the nature of SOEs. It is of great practical and theoretical significance to make a more comprehensive and accurate judgment on the efficiency of SOEs. This paper constructs an index of local governments' implicit contingent liabilities in 31 provinces based on the 488 local SOEs to study the impact of implicit contingent liabilities, and the time period is the year 2007 to the year 2020. Our findings show that an increase in local SOEs' assets suppresses economic fluctuations at the cost of increasing government's implicit contingent debt and has a negative impact on economic growth. Unlike the fiscal influence path of explicit debt, implicit contingent debt restrains local economic growth through financial markets. The deleveraging of local SOEs and improving their efficiency can improve the overall efficiency of local funds and reduce the negative effect of local governments' implicit contingent liabilities on economic growth.

**Keywords:** efficiency of local SOEs; implicit contingent liabilities; balance between the leverage and growth; leverage transfer; efficiency of capital

## 1. Introduction

In 2008, the Chinese government put forth an economic stimulus package of 586 billion USD to minimize the impact of the global financial crisis. Driven by the policy, an increase in SOEs' investments maintained the Chinese economic growth and calmed the economic fluctuation during the financial crisis [1]. At the same time, the high leverage ratio of the public sector, which is composed of local governments and SOEs in China, has become the "gray rhinoceros" in China's systemic financial risks and threatens economic growth [2]. After the financial crisis, compared with stabilizing the fluctuation of the economy, promoting economic growth has become a more important goal of the governments. However, the previous model of relying on increasing leverage to promote economic growth is no longer applicable. Therefore, under the current economic background of China, it is of great significance to study how to promote economic growth without increasing the total social leverage and realize the dynamic balance between maintaining leverage and promoting growth [3]. Liu Xiaoguang et al. [4] believes that the efficiency of leverage is crucial to economic growth.

In terms of the structure, the leverage of China is higher than emerging countries but lower than developed countries, so the overall leverage risk is controllable. However, the leverage of non-financial enterprises is far higher than the world average, among which, the debt of SOEs accounts for more than 60% [3]. SOEs are important policy instruments for local governments for achieving high-quality economic development by taking on some policy responsibilities, such as counter-cyclically investing in times of crisis [5]. The status of SOEs in China is similar to the "too big to fail" status of large financial institutions in Western countries [6,7], and governments provide guarantees and assistance to SOEs. When the economic growth decreases, stimulus policies often lead to a deviation in asset scale and capital efficiency of local SOEs and increase the probability of default of enterprises. In order to reduce the default risk of local SOEs, governments provide guarantees and rescue promises to them, forming implicit contingent liabilities of local governments and aggravating the local finance risks. Governments' debts caused by SOEs are not included in the statistics of governments' direct debt, and whether governments need to repay the debts depends on whether the SOEs are facing the default risks. According to Hana [8], Handayani and Damayanti [9] and Soler and Sy [10], local government liabilities formed by local SOEs are implicit contingent liabilities of governments, which is the focus of high leverage risks in China at present.

Local governments are responsible for public risks, which include the liabilities incurred beyond the provisions of relevant laws and contracts. The payment of these liabilities depends on the occurrence of a specific event. Hence, these liabilities are known as implicit contingent liabilities [8,11,12]. The implicit contingent liabilities of local governments in China can be divided into three categories: constructive debts, consumer debts and the debts due to local governments' financing guarantees. Among the sources of implicit contingent liabilities of local governments in China, scholars' research mainly focuses on the financing platform of local governments, which is a special type of SOE. Bai et al. [13] thinks that financing platform debt is also local governments' debt in essence, and Hu Wenxiu and Zhang Xinxing [14] built an improved KMV model to measure the pressure of local governments' implicit contingent liabilities formed by financing platforms. Due to a series of central government policies aimed to regulate the financing behavior of local governments, financing platforms as the main sources of implicit contingent liabilities of local governments have been set up by government departments, while there is little research on the financial risks that may be induced by other SOEs. The data set on governments' contingent liabilities constructed by Bova et al. [15] shows that the ratio of SOEs' government expenditure to GDP in 1996 and 2003 was 0.3% and 0.1%, respectively, which indicates a high fiscal cost, and that SOEs other than financing platform are also important sources of implicit contingent liabilities of local governments. Therefore, this study focuses on local governments' implicit contingent liabilities from local SOEs and analyzes their influence on the economy and the influence mechanism.

## 2. Literature Review

There is no consensus on the economic impact of local governments' implicit debts. Firstly, considering that SOEs are responsible for the implicit contingent liabilities, we review the literature of the SOEs. SOEs, as the sources of the liabilities, can help stabilize economic growth, raise macroeconomic efficiency and achieve the optimal allocation of overall social resources [16]. In terms of macroeconomic efficiency, during the financial crisis in 2008 and the Great Depression in 1930s, many countries set up SOEs to stabilize the economy, and SOEs are effective instruments to achieve some governments' objectives during crisis periods [17–21]. Wang Wencheng [22], Zhan Xinyu and Fang Fuqian [23] and Guo Jing and Ma Guangrong [2] also found that the investment of SOEs can stabilize economic growth. However, over-investments by SOEs have a "crowding out effect" on the investments of private enterprises [24]. If the operating efficiency and profitability of SOEs are less than that of private enterprises, the excess liabilities of SOEs will re-

duce their contribution to overall growth [25–29]; that is, over-investment by SOEs could reduce growth.

Suppressing economic fluctuations is the macroeconomic efficiency of SOEs [1], while reducing the growth is the microeconomic efficiency of SOEs [27]. So, we can have a more complete understanding of the SOEs by combining macroeconomic efficiency with microeconomic efficiency. SOEs are facing pressure of payment because of low microeconomic efficiency, which would increase the risks of default and induce local governments' implicit contingent liabilities [30]. The implicit contingent liabilities reflect the microeconomic efficiency of SOEs, so studying the impact of them on the economic growth combines the macro with microeconomic efficiency, which is helpful to make the status of SOEs clear. The investment of SOEs and non-SOEs to promote economic growth can be regarded as the allocation and combination of capital between them, which is similar to the mean-variance model. The balance between macroefficiency and microefficiency of SOEs is similar to maximizing profits while minimizing risks in the mean-variance model. This paper analyzes the macroefficiency and microefficiency of SOEs and the ways to balance them in terms of implicit contingent liabilities incurred by SOEs. Based on this, the more efficient mean-variance portfolio selection model proposed by Dai and Kang et al. [31] can be referenced in future studies to further research the optimal leverage allocation. Financial risk measurement is the core link to financial market management [32]. Therefore, we first construct an index of implicit contingent liabilities incurred by local SOEs.

Implicit contingent liabilities incurred by SOEs are important components of government debt systems, but the impact mechanisms of governments' implicit contingent debts and explicit debts on economic growth are different. There is no consensus on the economic impact of local governments' explicit debts, including inverted U-shaped influence, U-shaped influence and negative influence, etc. [33–36]. Explicit debts generally affect economic growth through fiscal and monetary channels, such as fiscal deficit, tax rate, and changes in long-term interest rate and inflation [37–40]. Implicit contingent liabilities are a potential pressure point for local governments, since they do not have to be paid before explicit, so it is difficult for governments to adjust their current financial arrangements beforehand. In summary, we study whether the implicit contingent debts of local governments originating from SOEs are "a blessing or a curse", and its relationships with economic growth.

We construct an index to measure local governments' implicit contingent liabilities based on SOE data [41] and study the influencing mechanism of SOEs' macroefficiency and microinefficiency on the economy. The results show that an increase in local SOEs' investments suppresses economic fluctuations but increases local governments' implicit contingent debts, which has a negative impact on economic growth. Unlike the fiscal influence of explicit debts, implicit contingent debts from SOEs block the virtuous cycle of credit and indirectly restrain local economic growth. The deleveraging of local SOEs and shareholding reform can improve the overall efficiency of local funds and reduce the negative effect of implicit contingent liabilities of local governments on economic growth.

The contributions of this study are as follows. First, most of the literature on implicit contingent liabilities of government mainly focus on the national government at present. This paper focuses on the implicit contingent liabilities of provincial local governments. Second, most the literature on the implicit debt of local governments studies the implicit debt of local governments incurred by the financing platform of local governments, which is a special type of local SOE, and less attention is paid to other SOEs. This paper takes these SOEs as the object to study the implicit contingent debt of local governments. Third, this paper combines the macroefficiency with the microefficiency of SOEs through local governments' implicit contingent liabilities and comprehensively analyzes their status and role in the economy from two aspects: the counter-cyclical investment of SOEs to suppress economic fluctuations and low microefficiency to reduce economic growth.

The remainder of this paper is organized as follows. Section 3 theoretically analyzes the influence of the leverage of SOEs on economic growth. Section 4 introduces the data, variables and the model setting. Section 5 presents the empirical results and discussions. Section 6 presents the results of robustness tests. Section 7 summarizes the main findings and suggestions.

### 3. Theoretical Framework

#### 3.1. The Measurement Index of the Implicit Contingent Liabilities

SOEs in China are similar to large financial institutions in Western countries, which are considered "too big to fail" [42,43]. Therefore, we used Arslanalp and Liao's [41] method to construct an index to measure the governments' implicit contingent liabilities incurred by local SOEs.

Firstly, Equation (1) calculates the expectations of local government debts from one SOE. $TAL_{it}$ is the total debts of SOE $i$ in year $t$. $PD_{it}$ means the probability of default for enterprise $i$ in year $t$, calculated using the KMV model. $LGD$ denotes the loss ratio given that the corporation defaults and $PSS$ is the probability of the enterprises being rescued by local governments. As we chose to focus on the relationship between implicit contingent liabilities and economic growth, not the absolute scale of debts, we set the value of $LGD$ at 50% (we also set it at 40% and 60% and obtained similar results). According to the special status of SOEs in China, $PSS$ is set to 1.

$$EL_{it} = TAL_{it} * PD_{it} * LGD * PSS_{it} \qquad (1)$$

Secondly, Equation (2) calculates the expectations of local government debts formed by all local SOEs in the same province by summing them up. Further, because the local SOEs are mainly in traditional industries, the default risks among them are highly related. Therefore, Equation (3) calculates the governments' unexpected debt pressure caused by the high relation among the local SOEs.

$$EL_t = \sum_i EL_{i,t} \qquad (2)$$

$$UL_t = 2 * sqrt\left(Var\left(\sum_i^n EL_{i,t}\right)\right) = 2\sqrt{TAL_{i,t} * PSS_{i,t} * LGD * \sum PD * TAL_{i,t} * PSS_{i,t} * LGD'} \qquad (3)$$

$$\sum_{PD}(i,j) = PD_{ij} - PD_i * PD_j \qquad (4)$$

The sum of expected government debts and unexpected government debts incurred by local SOEs forms the implicit contingent liabilities of local governments, as shown in Equation (5). Finally, to exclude the impact of differences in the level of economic development on the index, the results of Equation (5) are standardized with local GDP to obtain the implicit contingent liability index (*CLI*) of local governments.

$$CLI_t = EL_t + UL_t \qquad (5)$$

In this paper, except for the calculation of the index, which requires the data of the each SOEs, other data related to non-SOEs and SOEs are all overall statistical data published on the official website of the National Bureau of Statistics of China. The data used in index calculation are from the annual reports of each company and the stock market statistics in the Wind database.

#### 3.2. Theoretical Framework of Emipirical Study

At present, under the guidance of neoclassical economic theory, most of the research on the efficiency of SOEs use microefficiency indicators such as "financial index", "loss index" and "total factor productivity index" to evaluate efficiency [44,45], and form a

consensus that SOEs have low microefficiency. However, many classic socialist theorists considered that macroeconomic efficiency is important for a socialist economic system. The classical socialist theories and the economic stability in each financial crisis of China show that the macroeconomic efficiency of SOEs is more important than the microeconomic efficiency, and microeconomic efficiency in neoclassical economic theory cannot reflect the nature of SOEs [16,46]. In socialist theory, SOEs are the coordinators to overcome and coordinate "market failure" and "government failure" [47]. On the one hand, by the reformation of corporations, SOEs should accept the market constraints and incentives as the main part of market competition; on the other hand, governments indirectly control SOEs with financial subsidies, guarantees and rescue promises and regards them as the transmission mechanism of government macro-control. Therefore, it is of great practical and theoretical significance to make a more comprehensive and accurate judgment on the efficiency of SOEs.

### 3.2.1. The Formation of Local Governments' Implicit Contingent Liabilities

Local SOEs help governments achieve policy goals [16,20,21,46,48], such as maintaining the high-quality development of China's economy. During the financial crisis, the counter-cyclical investments of SOEs acted as a "macroeconomic stabilizer" [1]. In 2009, the investments of SOEs increased by about 2.1 trillion yuan, while that of non-SOEs increased by only about 1.1 trillion yuan, half of what the SOEs invested. There were significant differences in the investments of SOEs and non-SOEs in China after the financial crisis [49]. The pro-cyclicality of non-SOEs' investments led to a sharp decline in investment during the crisis and exacerbated the economic fluctuations. The counter-cyclicality of SOEs' investments led to an increase in investment during the crisis and stabilized the macroeconomy by lowering economic fluctuations.

Many studies have confirmed that SOEs are low in efficiency [26,27,50,51]. To sustain economic growth, local governments intervene in local SOEs' investments by providing implicit guarantees to local SOEs [30]. Local SOEs counter-cyclically increase investments and rapidly increase their debts. However, their solvency may deteriorate because of the inefficient use of loans, thereby increasing their risk of default. With more governments' implicit guarantees being given, the implicit contingent liabilities of local governments increase. The implicit contingent liabilities of local governments are not only the results of governments' intervention in the market, but also a means for local governments to intervene in the market [52]. Based on the above analysis, we propose Hypothesis 1.

**Hypothesis 1 (H1).** *An increase in the asset scale of local SOEs reduces the economic fluctuations but also increases the implicit contingent liabilities of local governments.*

### 3.2.2. The Local Governments' Implicit Contingent Liabilities and Economic Growth

To maintain stable economic growth, local SOEs increase their scale of assets while simultaneously increasing the implicit contingent liabilities of local governments, which means that their debt scale and efficiency deviate [50]. Local SOEs' low efficiency in increasing debts cannot yield enough profits, so they have to borrow more to repay old debts and further increase their scale of debts to influence economic growth through non-SOEs and bank credits. However, capitals cannot form a virtuous circle because of their low efficiency, which leads to a tightening of debt constraints in the financial market [53,54]. Under this condition, SOEs can continue to obtain financing because of soft budget constraints [45,55,56], but non-SOEs are unable to obtain external financing and have to reduce their investments. Loans to local SOEs are sunk costs for the banks, and as long as the marginal revenue of providing loans to local SOEs is greater than the marginal cost of giving up, banks will continue to provide loans to local SOEs [57,58], which also reduces the available external financing for non-SOEs [24,59,60]. Further, the increasing investments of local SOEs exacerbates the overcapacity of non-SOEs [61], and non-SOEs reduce their investments on their own. Al-Janadi et al. [62] found SOEs' efficiency is lower



than that of non-SOEs, which reduces the overall efficiency of capitals. He and Kyaw [34], Uddin [27] and Liu [25] also consider that SOEs are not only inefficient themselves but also reduce the economic growth. Therefore, implicit contingent liabilities of local governments reduce the overall efficiency of capital and thus have a negative impact on economic growth.

At present, the financial system in China is still dominated by bank credit, which is the main channel for enterprises to obtain external funds [63]. However, bank credit is influenced by external factors and results in "credit discrimination". Credit discrimination exists in banks and other financial institutions in all countries [64–66]; non-SOEs in China are faced with the problem of "credit discrimination". SOEs are more likely to obtain bank credits compared with non-SOEs, and thus, local SOEs' loans account for a large part of banks' total credits because of "credit discrimination" [53,67]. So, it is important for banks to assess whether SOEs can repay loans on time. The low efficiency of local SOEs lowers their ability to repay debts due to an increase in debt scale and debt pressure, which in turn increases banks' problem loans. As problem loans increase, banks are forced to add to their reserves and reduce credits. Bank credit is an important financing source for the government to promote economic development [68–70]; an increase in implicit contingent liabilities of the government will inhibit the creation of bank liquidity, which will have a negative impact on the growth of real economy [71]. Based on the above analysis, we propose Hypothesis 2a and 2b.

**Hypothesis 2a (H2a).** *Implicit contingent liabilities have a negative impact on economic growth.*

**Hypothesis 2b (H2b).** *Implicit contingent liabilities reduce economic growth through the total efficiency of capitals and banks' credits.*

3.2.3. How to Balance between Implicit Contingent Liabilities and Economic Growth

The implicit contingent liabilities of local governments incurred by local SOEs are a key factor affecting economic growth. Deleveraging can reduce its negative effects on economic growth. However, some studies have found that a sharp reduction in leverage could reduce economic growth and may even lead to long-term recession [72–76]. Liu Xiaoguang et al. [4] believed that the efficiency of leverage is crucial to economic growth and at the core of striking a balance between leverage and economic growth.

A transfer of leverage from SOEs to non-SOEs can improve capital efficiency. Sun Xiaohua and Li Mingshan [51], Ji Min et al. [45] and Xu Zhaoyuan and Zhang Wenkui [44] believe that the capital efficiency of non-SOEs is higher than that of SOEs, so a transfer can improve the overall efficiency of capital. Further, the high efficiency of non-SOEs means that they can repay debts on time and can reduce the problem loans of banks. So, banks can offer more credits and promote economic growth.

Although a transfer of leverage from SOEs to non-SOEs can improve the overall efficiency of capital and promote economic growth, this positive effect is limited. The transfer to non-SOEs increases the liabilities of private enterprises and reduces their efficiency of capital, thereby increasing their risk of bankruptcy [45]. Hence, improving the capital efficiency of SOEs through reform is a more effective way of promoting economic growth. Based on the above analysis, we propose Hypothesis 3a and 3b.

**Hypothesis 3a (H3a).** *A transfer of leverage from SOEs to non-SOEs improves the enterprises' efficiency and can relieve the negative effects of implicit contingent liabilities on economic growth.*

**Hypothesis 3b (H3b).** *There is an inverted U-shaped relationship between the non-SOEs' capital proportion and the overall capital efficiency; thus, transfer has a limited positive effect.*

## 4. Empirical Model and Variable Specification

### 4.1. Data Sources and Data Processing

We selected all listed local SOEs established before 2002 to exclude the impact of new companies' unusual data, and the time interval we studied was from 2007 to 2020. The data sources were the Wind database and National Bureau of Statistics. We selected listed companies because: (1) there is more detailed financial information about them and (2) listed companies are generally the leading enterprises in a local area and are crucial for local economic development. Hence, local governments are more willing to provide guarantees and assistances to them. Besides, the financial performance of listed companies is better than that of others, so the index of implicit contingent liabilities means a better situation when we use the data of listed local SOEs. The situation would be worse if unlisted local SOEs were included in the sample.

To ensure data quality, we excluded financial companies, as the accounting standards for financial companies are different from those of non-financial enterprises, and there is no comparability between the two. Some companies that have incomplete data were also excluded. Moreover, companies marked as ST (meaning that this company is delisted) were also excluded, because the data of them are abnormal. After data screening, there were 488 listed local SOEs in the sample. Even though this number is less than 1% of all local SOEs, their capital base is large. So, this was a representative sample.

The data of local SOEs were updated to the year 2020, and thus, implicit contingent liabilities of local governments calculated with these data were also updated to the year 2020. However, according to the calculation method of variable "stdgdp", the time range of the empirical study had to be reduced by 2 years to 2018. In addition, some statistic data on the official website of the National Bureau of Statistics of China are only updated to the year 2017, so the time range of the follow-up empirical study will be from 2007 to 2017. Accordingly, panel data of 341 research samples in 31 provinces in 11 years were finally formed. Table 1 shows the provincial distribution of the local SOEs in the sample. We found that there is less than 10 local SOEs each in GuiZhou, HaiNan, HeiLongJiang, JiLin, Inner Mongolia, NingXia, QingHai, Tibet and ChongQing, all economically backward areas. Since the total number of local SOEs in these provinces is also low, these companies have a significant effect on local economic growth. So, these provinces were included in the study.

**Table 1.** Provincial distribution of the local SOEs.

| Province | Numbers | Province | Numbers | Province | Numbers |
|---|---|---|---|---|---|
| AnHui | 26 | BeiJing | 26 | FuJian | 23 |
| GanSu | 8 | GuangDong | 44 | GuangXi | 12 |
| GuiZhou | 5 | HaiNan | 4 | HeBei | 12 |
| HeNan | 14 | HeiLongJiang | 6 | HuBei | 16 |
| Hunan | 17 | JiLin | 7 | JiangSu | 28 |
| JiangXi | 12 | LiaoNing | 16 | Inner Mongolia | 2 |
| NingXia | 1 | QingHai | 3 | ShanDong | 35 |
| ShanXi | 16 | ShanXi | 13 | ShangHai | 56 |
| SiChuan | 17 | TianJin | 14 | Tibet | 3 |
| XinJiang | 10 | YunNan | 10 | ZheJiang | 25 |
| ChongQing | 6 | | | | |

*4.2. Variable Description*

According to the three objects of our study, there are three dependent variables: the standard deviation of GDP growth (stdgdp), the growth rate of GDP (dgdp) and the efficiency of total capital (ttm). Stdgdp represents the economic fluctuation, and dgdp represents the economic growth.

The main independent variable of this paper is the implicit contingent liability index of local governments calculated by Equations (1)–(5). Figure 1 shows the average implicit contingent liabilities of local governments incurred by local SOEs in the eastern, central and western regions and for the whole country. Implicit contingent liabilities grew rapidly after 2008. Although their growth rate declined around 2010, it started growing rapidly again thereafter. The financial crisis of 2008 increased the default risks of local SOEs and led to a rapid increase in implicit contingent liabilities. Expansionary policies were put in place during the crisis to sustain economic growth, and these led to a slowing down in the growth of implicit contingent liabilities. However, once the overcapacity of SOEs and the risks of local governments' debts became apparent, the central government gradually moved away from expansionary policies to contractionary policies. This increased the financial pressure on SOEs, and the implicit contingent liabilities of local governments increased again. This trend of implicit contingent liabilities of local governments is shown in Figure 1, and the index of implicit contingent liabilities is appropriate.

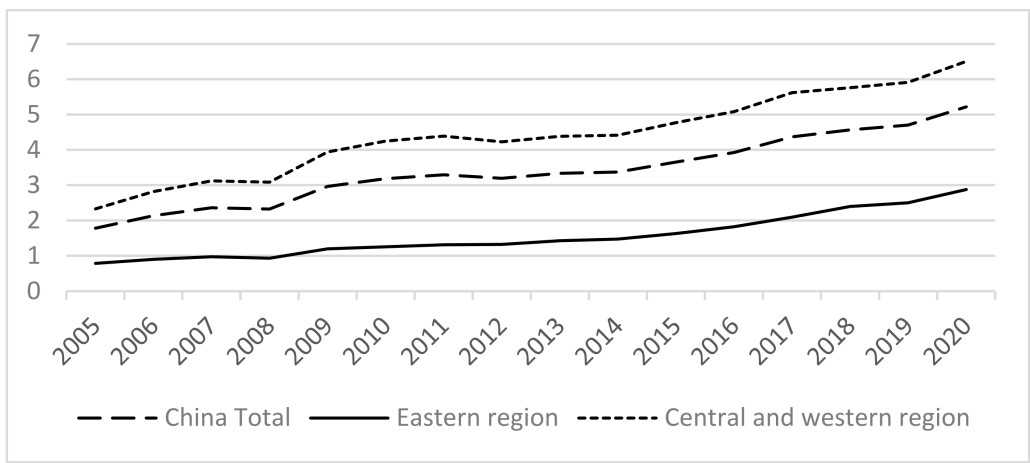

**Figure 1.** The trend of the implicit contingent liability. Source: constructed in this paper.

In the analysis of the influence of local governments' implicit contingent liabilities on economic growth, there are several intervening variables: the increase rate of problem loans (dsbad) and the efficiency of total capital (ttm). In the analysis on how to balance between leverage and economic growth, we paid attention to the effects of the proportion of SOEs' assets (a_rstate), the proportion of non-SOEs' assets (a_rpri), the efficiency of SOEs (statettm) and the efficiency of non-SOEs (pri_state) on the efficiency of total capital in each province (ttm).

Economic growth is affected by other factors too. We included the effects of governments in the economy (gov), human capital (edu), foreign direct investments (dfdi), investment rate (inv), and urbanization rate (city) as the control variables according to Rui-ming Liu [77]. Further, local economic growth and implicit contingent liabilities of local governments could be influenced by identical variables, such as policy shocks, so we also included environmental policies (lnpo), anti-corruption policy (corruption), and the government policy of local governments' debt (policy) in the empirical model. We also included a time dummy variable to identify the financial crisis in 2008 (crisis).

The calculation method and meanings of variables are shown in Table 2.

**Table 2.** Variable descriptions.

| | Variable Name | Variable Symbol | Variable Descriptions and Calculations | Reference |
|---|---|---|---|---|
| Dependent Variables | Standard deviation of economic growth | stdgdp | The degree of economic fluctuation in year t is expressed by the year-on-year standard deviation of GDP in a 5-year window period from year (t − 2) to year (t + 2) | Guo Jing and Ma Guangrong [1] |
| | Economic growth rate | dgdp | Annual economic growth rate in each province | Rui-ming Liu [77] |
| | Total efficiency of money | ttm | Proportion of SOEs' assets × efficiency of SOEs + proportion of private enterprises' assets × efficiency of private enterprises | Guo Jiangshan and Li Zixuan [78] |
| | | ttd | Replace the assets by debts, and the method is identical with ttm | |
| Independent Variables | Changes in local governments' implicit contingent liabilities | rcli | Rate of the implicit contingent liability index of local governments, which is calculated by Equations (1)–(5), | Arslanalp S and Liao Y [41] |
| | The increase in cli | dcli | First-order difference of $cli$, $cli_{i,t+1} - cli_{i,t}$ | |
| | Instrument variables of rcli | rcli_iv | The index of one province's implicit contingent liabilities adopts the average index of implicit contingent liabilities of the other 30 provinces as an instrument variable. | Checherita-Westphal and Rother [79] |
| | Proportion of enterprises' assets | m_rstate | Asset of SOE/(asset of SOE + asset of POE) | Guo Jing and Ma Guangrong [1] |
| | | m_rpri | Asset of non-SOEs/(asset of SOE + asset of POE) | |
| | Proportion of enterprises' debts | d_rstate | Debt of SOE/(debt of SOE + debt of POE) | |
| | | d_rpri | Debt of non-SOEs/(debt of SOE + debt of POE) | |
| | Enterprises' efficiency of money | statettm | Prime operating revenue of SOE/asset of SOE | Guo Jiangshan and Li Zixuan [78] |
| | | prittm | Prime operating revenue of non-SOEs/asset of non-SOEs | |
| | | statettd | Prime operating revenue of SOE/debt of SOE | |
| | | prittd | Prime operating revenue of non-SOEs/debt of non-SOEs | |
| | Increase rate of problem loans | dsbad | Increase rate of problem loans | Guo et al. [53] |
| | Region dummy variables | region | Central and western region = 1; eastern region = 0 | - |
| Control Variables 1 | Governments' role in the market | gov | Government spending in each province/GDP in each province | Guo Jing and Ma Guangrong [1] |
| | Human capital | edu | College students' enrollments in each province/population in each province | Rui-ming Liu [77] |
| | Foreign direct investment | dfdi | Increase rate of FDI | Guo Jing and Ma Guangrong [1] |
| | Investment rate | inv | Investments/GDP | Rui-ming Liu [77] |
| | Proportion of urban population | city | Populations in city/populations in each province | |
| Control Variables 2 | Environment protection policy | lnpo | Investments in environmental pollution control | He Jue [80] |
| | Anti-corruption policy | Corruption | Corruption governance indicators in WGI index (Worldwide Governance Indicators) published by the WB (World Bank) | Wang Maobin and Kong Dongmin [81] |
| | Management policy of local governments' debts | policy | Take the No.43 policy as the representative, and build the dummy variables | Mao Jie and Cao Jing [82] |
| | Financial crisis | crisis | Financial crisis period is from 2008 to 2009, and we built the dummy variables | Arslanalp S and Liao Y [41] |

Notes: Control Variables 1 includes some macroeconomic variables; Control Variables 2 includes some policy variables.

### 4.3. Model Specification and Data Statistics

First, we studied the effects of implicit contingent liabilities on economic fluctuation using Equation (6). The expected result was $\beta_1 < 0$, which means that raising the asset proportion of local SOEs reduces the volatility of economic growth. Then, we replaced the stdgdp with cli in Equation (6) and studied the relationship between asset proportion of local SOEs and implicit contingent liabilities of local governments. The expected result was $\beta_1 > 0$, indicating that raising the asset proportion of local SOEs increases implicit contingent liabilities. We found that local SOEs can help reduce the volatility of economic growth at the cost of implicit contingent liabilities.

$$\text{stdgdp}_{i,t} = \beta_0 + \beta_1 * m\_rstate_{i,t} + \gamma * controls_{i,t} + \epsilon_i + \varepsilon_{i,t} \tag{6}$$

Second, we studied the relationship between implicit contingent liabilities and economic growth using Equation (7). The expected result was $\beta_1 < 0$, which means that implicit contingent liabilities will depress economic growth. After this, we used the mediating effect Equations (8)–(10) to study the influence channels on the economy. First, we studied the role of the rate of increase in problem loans (dsbad). Equation (8) is the same as Equation (7). Equation (9) shows the relationship between the rate of increase in implicit contingent liabilities (rcli) and the rate of increase in problem loans (dsbad). The dependent variable in Equation (10) is the growth rate of GDP (dgdp), and the main independent variables are the rate of increase in implicit contingent liabilities (rcli) and the rate of increase in problem loans (dsbad). We also replaced the rate of increase in problem loans (dsbad) with the efficiency of total capital calculated with the asset data (ttm) and debt data (ttd) in Equations (9) and (10) to study the role of these two variables. According to Wen Zhonglin and Ye Baojuan [83], if the coefficients c, a and b are significantly different from 0, it shows a significant mediating effect. If c' is significantly different from 0, it shows an incomplete mediating effect, and if c' is not significantly different from 0, it shows a complete mediating effect. If at least one of c, a and b is not significantly different from 0, further testing is needed, we used the bootstrap method for that.

$$\text{Dgdp}_{i,t} = \beta_0 + \beta_1 * rcli_{i,t} + \gamma * controls_{i,t} + \epsilon_i + \varepsilon_{i,t} \tag{7}$$

$$\text{dgdp}_{i,t} = \beta_0 + c * rcli_{i,t} + \gamma * controls_{i,t} + \epsilon_i + \varepsilon_{i,t} \tag{8}$$

$$\text{dsabd}_{i,t} = \beta_0 + a * rcli_{i,t} + \gamma * controls_{i,t} + \epsilon_i + \varepsilon_{i,t} \tag{9}$$

$$dgdp_{i,t} = \beta_0 + c' * rcli_{i,t} + b * dsbad_{i,t} + \gamma * control_{i,t} + \epsilon_i + \varepsilon_{i,t} \tag{10}$$

Third, we further studied the ways of balancing leverage and economic growth. Equation (11) first studies the influence of the asset proportion of SOEs (a_rstate) on the efficiency of total capitals in each province (ttm). Then, we replaced the asset proportion of SOEs (a_rstate) by the asset proportion of non-SOEs (a_rpri), the efficiency of SOEs (statettm) and the efficiency of non-SOEs (prittm) to study the influence of these three variables. $\beta_1 > 0$ means that the main independent variable raises the efficiency of total capitals in each province (ttm) and $\beta_1 < 0$ means that the main independent variable reduces the efficiency of total capitals in each province (ttm). Equation (12) studies the non-linear relationship between the efficiency of private enterprises and the proportion of private enterprises' assets (m_rpri).

$$ttm_{i,t} = \beta_0 + \beta_1 * m\_rstate_{i,t} + \gamma * controls_{i,t} + \epsilon_i + \varepsilon_{i,t} \tag{11}$$

$$pri\_ttm_{i,t} = \beta_0 + \beta_1 * rpri2_{i,t} + \beta_2 * rpri_{i,t} + \gamma * controls_{i,t} + \epsilon_i + \varepsilon_{i,t} \tag{12}$$

In Equations (6)–(12), the coefficient '$\gamma$' represents the coefficient matrix of all control variables, '$\epsilon_i + \varepsilon_{i,t}$' is the composite error term, '$\epsilon_i$' is the intercept term of individual heterogeneity and '$\varepsilon_{i,t}$' is the disturbance term that varies with individuals and time. Table 3 represents the statistics of all the above-mentioned variables.

**Table 3.** Descriptive statistics.

| Variables | Observations | Average | Median | Standard Deviation | Minimum | Maximum |
|-----------|--------------|---------|--------|--------------------|---------|---------|
| stdgdp | 341 | 0.0149 | 0.0136 | 0.0084 | 0.0007 | 0.0487 |
| dgdp | 341 | 0.1313 | 0.1195 | 0.0716 | −0.2240 | 0.3227 |
| rcli | 341 | 0.0899 | 0.0570 | 0.3599 | −0.7806 | 6.1536 |
| cli | 341 | 3.705 | 1.3632 | 7.6134 | 0.0001 | 50.5189 |
| dcli | 341 | 0.2027 | 0.0623 | 0.6635 | −2.9019 | 6.8883 |
| rcli_iv | 341 | 0.0702 | 0.0729 | 0.0807 | −0.0587 | 0.3259 |
| gov | 341 | 0.3001 | 0.2794 | 0.0849 | 0.1895 | 0.6541 |
| dfdi | 341 | 0.1813 | 0.1173 | 0.4560 | −0.7134 | 6.9747 |
| edu | 341 | 0.0241 | 0.0222 | 0.0094 | 0.0090 | 0.0683 |
| inv | 341 | 0.7344 | 0.7336 | 0.2418 | 0.2366 | 1.5070 |
| city | 341 | 0.5349 | 0.5178 | 0.1434 | 0.2150 | 0.8960 |
| corruption | 341 | −0.4209 | −0.44 | 0.1193 | −0.5900 | −0.2500 |
| lnpo | 341 | 8.8452 | 9.0184 | 0.3484 | 8.1278 | 9.1670 |
| dsbad | 341 | 0.1114 | 0.0418 | 0.4316 | −0.8614 | 2.3913 |
| ttm | 341 | 0.9365 | 0.9376 | 0.3771 | 0.1194 | 1.9022 |
| ttd | 341 | 1.6303 | 1.6221 | 0.6879 | 0.2345 | 3.5278 |
| statettm | 341 | 0.6901 | 0.7108 | 0.2207 | 0.0959 | 1.1995 |
| prittm | 341 | 1.4961 | 1.5723 | 0.6314 | 0.2949 | 3.0722 |
| statettd | 341 | 1.1483 | 1.1629 | 0.3755 | 0.1963 | 2.1374 |
| prittd | 341 | 3.0165 | 2.8820 | 1.6525 | 0.6193 | 9.0214 |
| a_rstate | 341 | 0.7320 | 0.7751 | 0.1620 | 0.2541 | 0.9603 |
| a_rpri | 341 | 0.2680 | 0.2249 | 0.1620 | 0.0397 | 0.7459 |
| d_rstate | 341 | 0.7592 | 0.7922 | 0.1505 | 0.2316 | 0.9640 |
| d_rpri | 341 | 0.2408 | 0.2078 | 0.1505 | 0.0360 | 0.7684 |

Sources: Wind.

## 5. Empirical Research Results

### 5.1. The Implicit Contingent Liabilities and the Economic Fluctuations

Table 4 reports the relationship among local SOEs' assets, the economic fluctuations and the implicit contingent liabilities of local governments. Columns (1) and (2) of the table show the relationship between the proportion of local SOEs' assets and the implicit contingent liabilities. The results show that the proportion of local SOEs' assets is positively related with the implicit contingent liabilities. Columns (3)–(5) show the relationship between the implicit contingent liabilities and the economic fluctuations, which point to the macroefficiency of local SOEs. Column (3) contains no control variables. Column (4) contains macroeconomic control variables and column (5) not only contains macroeconomic control variables, but also contains the policy shocks. We mainly focus on the results of column (4) and (5). The implicit contingent liabilities are negatively related with the fluctuations of local economic growth, which means that local SOEs' investments can suppress economic fluctuations. The results of Table 4 show that an asset expansion of local SOEs can calm fluctuations of local economic growth but increases implicit contingent liabilities. The results of Table 4 support hypothesis 1.

**Table 4.** Local SOEs reduce economic growth fluctuations.

| | Cli | | stdgdp | | |
|---|---|---|---|---|---|
| | **(1)** | **(2)** | **(3)** | **(4)** | **(5)** |
| cli | | | −0.000866 * | −0.000372 ** | −0.000461 ** |
| | | | (0.000436) | (0.000180) | (0.000171) |
| a_rstate | 9.810 ** | 11.13 ** | | | |
| | (4.346) | (5.125) | | | |
| gov | 1.458 | 0.521 | | −0.0267 | −0.0522 * |
| | (2.560) | (1.974) | | (0.0318) | (0.0299) |
| dfdi | 0.0265 | 0.127 | | −0.000279 | 0.000435 |
| | (0.106) | (0.167) | | (0.000478) | (0.000382) |
| edu | −52.88 * | −11.72 | | 1.036 *** | 1.054 ** |
| | (27.84) | (43.26) | | (0.377) | (0.398) |
| inv | 3.616 * | 3.505 * | | −0.00451 | −0.00247 |
| | (1.861) | (1.793) | | (0.00704) | (0.00821) |
| city | 10.06 *** | −1.469 | | −0.0598 ** | −0.0743 * |
| | (3.483) | (7.477) | | (0.0278) | (0.0414) |
| corruption | | 0.797 | | | 0.0107 |
| | | (0.629) | | | (0.00920) |
| lnpo | | 1.172 | | | 0.00496 |
| | | (0.865) | | | (0.00323) |
| policy | | 0.125 | | | −0.00754 *** |
| | | (0.0933) | | | (0.00117) |
| crisis | | −0.00375 | | | −0.00211 ** |
| | | (0.0690) | | | (0.000936) |
| Constant | −11.12 * | −16.64 | 0.0178 *** | 0.0346 ** | 0.0119 |
| | (5.696) | (10.27) | (0.00142) | (0.0135) | (0.0169) |
| Observations | 341 | 341 | 341 | 341 | 341 |
| R-squared | 0.296 | 0.327 | 0.025 | 0.112 | 0.249 |
| Number of id | 31 | 31 | 31 | 31 | 31 |

Notes: 1. The figures in parentheses show the cluster robust standard errors; 2. *** indicates that the statistic is significant at the 1% level, ** indicates that the statistic is significant at the 5% level and * indicates that the statistic is significant at the 10% level.

### 5.2. The Implicit Contingent Liabilities and the Economic Growth

Table 5 reports the influence of implicit contingent liabilities on economic growth. Column (1) contains no control variables. Column (2) contains the macroeconomic variables. Column (3) contains the macroeconomic and the policy variables. Column (4) contains macroeconomic variables and the time dummy variable. The table shows that implicit contingent liabilities of local governments reduce economic growth. Column (5) joins the region dummy variables. The implicit contingent liabilities are negatively related with economic growth, and the coefficient of the cross term between the implicit contingent liabilities and region dummy variables is positive, which means there is less negative impact on economic growth in the central and western regions.

**Table 5.** Implicit contingent liabilities of local governments and economic growth.

| | Dgdp | | | | |
|---|---|---|---|---|---|
| | **(1)** | **(2)** | **(3)** | **(4)** | **(5)** |
| rcli | −0.0321 *** | −0.0212 ** | −0.0190 *** | −0.0116 ** | −0.203 *** |
| | (0.0109) | (0.00829) | (0.00710) | (0.00576) | (0.0393) |
| gov | | 0.568 *** | 0.469 *** | 0.340 *** | 0.451 *** |
| | | (0.149) | (0.128) | (0.102) | (0.124) |
| dfdi | | −0.00215 | −0.00635 | −0.00421 | −0.00646 |
| | | (0.00672) | (0.00580) | (0.00459) | (0.00561) |
| edu | | −0.506 | −4.470 *** | −1.659 | −4.445 *** |
| | | (1.731) | (1.575) | (1.328) | (1.521) |
| inv | | −0.0645 ** | −0.0350 | 0.0309 | −0.0448 * |
| | | (0.0300) | (0.0255) | (0.0213) | (0.0247) |
| city | | −0.666 *** | 0.505 ** | −0.158 | 0.544 *** |
| | | (0.137) | (0.201) | (0.195) | (0.194) |
| corruption | | | −0.538 *** | 0.255 | −0.518 *** |
| | | | (0.0684) | (0.330) | (0.0662) |
| lnpo | | | 0.00352 | 0.447 *** | −0.00187 |
| | | | (0.0172) | (0.141) | (0.0166) |
| policy | | | 0.00965 | −0.642 *** | 0.00858 |
| | | | (0.0112) | (0.135) | (0.0108) |
| crisis | | | −0.0195 ** | −0.255 *** | −0.0159 * |
| | | | (0.00875) | (0.0398) | (0.00849) |
| Region*rcli | | | | | 0.189 *** |
| | | | | | (0.0398) |
| time control variables | NO | NO | NO | YES | NO |
| Constant | 0.134 *** | 0.379 *** | −0.401 ** | −3.289 *** | −0.349 ** |
| | (0.00399) | (0.0759) | (0.178) | (1.201) | (0.172) |
| Observations | 341 | 341 | 341 | 341 | 341 |
| R-squared | 0.027 | 0.458 | 0.621 | 0.770 | 0.648 |
| Number of id | 31 | 31 | 31 | 31 | 31 |

Notes: 1. The figures in parentheses show the cluster robust standard errors; 2. *** indicates that the statistic is significant at the 1% level, ** indicates that the statistic is significant at the 5% level and * indicates that the statistic is significant at the 10% level.

Table 6 reports the influence mechanism of implicit contingent liabilities on economic growth. Column (1) is the result when the rate of increase in problem loans (dsbad) is the intervening variable. The result shows that implicit contingent liabilities and problem loans of banks are positively related, and they have a negative effect on economic growth. Column (2) and (3) are the results when the efficiency of total capital calculated with the asset data (ttm) and debt data (ttd) are intervening variables. The results show that implicit contingent liabilities and the efficiency of total capital are negatively related, and the efficiency of total capital have a positive effect on economic growth. Tables 5 and 6 support Hypothesis 2a and 2b.

**Table 6.** Mediating effect test.

| Mediating Effect Test | | | dgdp | | |
|---|---|---|---|---|---|
| | | | (1) | (2) | (3) |
| | | | dsbad | ttm | ttd |
| (1) | | c | −0.0190 *** | −0.0190 *** | −0.0190 *** |
| | | | (0.00710) | (0.00710) | (0.00710) |
| (2) | | a | 0.0912 * | −0.0302 * | −0.0658 * |
| | | | (0.0498) | (0.0161) | (0.0347) |
| (3) | | c′ | −0.0139 ** | −0.0171 ** | −0.0171 ** |
| | | | (0.00658) | (0.00707) | (0.00708) |
| | | b | −0.0556 *** | 0.0647 ** | 0.0292 ** |
| | | | (0.00758) | (0.0251) | (0.0117) |
| Control variables | | | Yes | Yes | Yes |
| Mediating effect | | | Yes | Yes | Yes |

Notes: 1. The figures in parentheses are the cluster robust standard errors; 2. *** indicates that the statistic is significant at the 1% level, ** indicates that the statistic is significant at the 5% level and * indicates that the statistic is significant at the 10% level. 3. This is the result of the Mediation Analysis. The coefficient 'c' shows the total effect of independent variable on dependent variable, the coefficient 'a' is the effect of variables 'rcli' on the variables 'dsbad'/'ttm'/'ttd', coefficient "b" is the effect of variables 'dsbad'/'ttm'/'ttd' on the variables 'dgdp', and coefficient "c′" is the effect of the variables 'rcli' on variables 'dgdp' after eliminating the influence of variables 'dsbad'/'ttm'/'ttd'.

### 5.3. How to Balance between Leverage and the Economic Growth

Table 7a,b show the results of how to balance between leverage and economic growth, and dependent variables and independent variables in Table 7a are in the same time, while the independent variables "a_rstate", "a_rpri","statettm"and "prittm" are lag variables in Table 7b. Column (1) and (2) in Table 7a show that the asset proportion of SOEs is negatively related with the efficiency of total capital, and the asset proportion of non-SOEs is positively related with the efficiency of total capital, so a transfer of leverage from SOEs to non-SOEs can improve the efficiency of total capitals and promote economic growth. Column (3) and (4) show that the efficiency of SOEs and non-SOEs has positive effects on the efficiency of total capitals, so an improvement in the efficiency of enterprises is an effective way to promote economic growth, by adjusting industrial structure, for example, to improve the enterprises' efficiency. The results also show that an improvement in SOEs' efficiency is more effective in improving the efficiency of total capital. Column (5) shows that there is an inverted U-shaped relationship between the non-SOEs' capital proportion and the overall capital efficiency.

**Table 7.** Ways to balance leverage and economic growth.

| (a) Ways to Balance Leverage and Economic Growth | | | | | |
|---|---|---|---|---|---|
| | | | ttm | | |
| | (1) | (2) | (3) | (4) | (5) |
| a_rpri*a_rpri | | | | | −1.265 ** |
| | | | | | (0.548) |
| a_rstate | −0.895 *** | | | | |
| | (0.158) | | | | |
| a_rpri | | 0.895 *** | | | 1.750 *** |
| | | (0.158) | | | (0.402) |
| statettm | | | 1.180 *** | | |
| | | | (0.0606) | | |

**Table 7.** *Cont.*

| | (a) Ways to Balance Leverage and Economic Growth | | | | |
|---|---|---|---|---|---|
| | **ttm** | | | | |
| | **(1)** | **(2)** | **(3)** | **(4)** | **(5)** |
| prittm | | | | 0.318 *** | |
| | | | | (0.0185) | |
| gov | 0.641 ** | 0.641 ** | −0.286 | 0.460 ** | 0.685 ** |
| | (0.278) | (0.278) | (0.199) | (0.207) | (0.276) |
| dfdi | 0.00296 | 0.00296 | 0.00484 | 0.00140 | 0.00376 |
| | (0.0127) | (0.0127) | (0.00882) | (0.00943) | (0.0126) |
| edu | 3.591 | 3.591 | 4.410 * | 1.059 | 4.091 |
| | (3.435) | (3.435) | (2.385) | (2.562) | (3.417) |
| inv | 0.0408 | 0.0408 | 0.138 *** | −0.0672 | 0.0178 |
| | (0.0558) | (0.0558) | (0.0389) | (0.0423) | (0.0563) |
| city | −0.264 | −0.264 | −0.343 | 0.758 ** | −0.259 |
| | (0.435) | (0.435) | (0.304) | (0.327) | (0.432) |
| corruption | −0.493 *** | −0.493 *** | 0.171 | −0.415 *** | −0.475 *** |
| | (0.149) | (0.149) | (0.110) | (0.111) | (0.148) |
| lnpo | 0.144 *** | 0.144 *** | 0.0632 ** | 0.104 *** | 0.133 *** |
| | (0.0379) | (0.0379) | (0.0267) | (0.0281) | (0.0379) |
| policy | −0.0742 *** | −0.0742 *** | −0.0106 | −0.0303 | −0.0714 *** |
| | (0.0244) | (0.0244) | (0.0174) | (0.0184) | (0.0242) |
| crisis | −0.0234 | −0.0234 | 0.00257 | 0.00669 | −0.0276 |
| | (0.0191) | (0.0191) | (0.0133) | (0.0142) | (0.0191) |
| Constant | −0.0297 | −0.924 ** | −0.300 | −1.147 *** | −0.931 ** |
| | (0.447) | (0.393) | (0.276) | (0.289) | (0.390) |
| Observations | 341 | 341 | 341 | 341 | 341 |
| R-squared | 0.363 | 0.363 | 0.689 | 0.645 | 0.374 |
| Number of id | 31 | 31 | 31 | 31 | 31 |
| | (b) Ways to Balance Leverage and Economic Growth with Lag Variables | | | | |
| | **ttm** | | | | |
| | **(1)** | **(2)** | **(3)** | **(4)** | |
| a_rstate | −0.663 *** | | | | |
| | (0.160) | | | | |
| a_rpri | | 0.663 *** | | | |
| | | (0.160) | | | |
| statettm | | | 0.829 *** | | |
| | | | (0.0591) | | |
| prittm | | | | 0.277 *** | |
| | | | | (0.0210) | |
| gov | 0.650 ** | 0.650 ** | 0.125 | 0.890 *** | |
| | (0.284) | (0.284) | (0.229) | (0.233) | |
| dfdi | 0.00694 | 0.00694 | 0.00959 | 0.00153 | |
| | (0.0130) | (0.0130) | (0.0103) | (0.0106) | |
| edu | 4.425 | 4.425 | 0.963 | −1.613 | |
| | (3.509) | (3.509) | (2.811) | (2.910) | |
| inv | 0.0705 | 0.0705 | 0.0387 | 0.0187 | |
| | (0.0567) | (0.0567) | (0.0454) | (0.0466) | |

**Table 7.** *Cont.*

| | (b) Ways to Balance Leverage and Economic Growth with Lag Variables | | | |
|---|---|---|---|---|
| | **ttm** | | | |
| | **(1)** | **(2)** | **(3)** | **(4)** |
| city | −0.351 (0.450) | −0.351 (0.450) | 0.983 *** (0.362) | 0.922 ** (0.370) |
| corruption | −0.463 *** (0.153) | −0.463 *** (0.153) | −0.611 *** (0.122) | −0.654 *** (0.125) |
| lnpo | 0.153 *** (0.0389) | 0.153 *** (0.0389) | 0.127 *** (0.0306) | 0.106 *** (0.0316) |
| policy | −0.0833 *** (0.0250) | −0.0833 *** (0.0250) | −0.0102 (0.0205) | −0.0341 * (0.0206) |
| crisis | −0.0190 (0.0195) | −0.0190 (0.0195) | −0.0196 (0.0156) | −0.0101 (0.0159) |
| Constant | −0.256 (0.469) | −0.919 ** (0.406) | −1.633 *** (0.318) | −1.413 *** (0.324) |
| Observations | 341 | 341 | 341 | 341 |
| R-squared | 0.333 | 0.333 | 0.574 | 0.554 |
| Number of id | 31 | 31 | 31 | 31 |

Notes: 1. The number in parentheses is the cluster robust standard error; 2.*** indicates that the statistic is significant at the 1% level, ** indicates that the statistic is significant at the 5% level and * indicates that the statistic is significant at the 10% level.

Column (1) and (2) in Table 7b show that the asset proportion of SOEs is negatively related with the efficiency of total capitals, and the asset proportion of non-SOEs is positively related with the efficiency of total capitals, so, the transfer of leverage from SOEs to non-SOEs can improve the efficiency of total capitals and promote economic growth. Column (3) and (4) show that the efficiencies of SOEs and non-SOEs have positive effects on the efficiency of total capitals, so an improvement of in the efficiency of enterprises is an effective way to promote economic growth. The coefficient of "statettm" is also larger than that of "prittm" in Table 7b; hence, an improvement in SOEs' efficiency is more effective. Table 7a,b support Hypothesis 3a and 3b.

## 6. Robustness Check

The above studies looked at the relationship between implicit contingent liabilities and economic growth and how to balance leverage and economic growth. We further tested the robustness of our results. Table 8 is the robustness test of the relationship between implicit contingent liabilities and economic growth. Column (1) replaces the previous independent variable "rcli" with "dcli", which is the first difference of "cli". Column (2) replaces the previous independent variable "rcli" with "rcli_iv", which is the instrumental variable of "rcli". Column (3)– (5) are the results of the instrumental variable regression (IV regression). Column (3) is the first stage result of the IV regression and shows that the instrument "rcli_iv" is highly correlated with the instrumented variable "rcli". In terms of the exogeneity, there is only one instrumented variable, and the number is same as the number of endogenous variables, so an overidentification test cannot be used here. However, this instrument has the advantage of not having a direct causal effect on the growth rate, at least if one assumes that there is no strong relationship between implicit contingent liabilities in other provinces and the GDP growth rate in one specific province. So, it is a valid instrument. Column (4) and (5) are the second stage results of IV regression with different estimating methods. The method of deviation transformation is proposed before the IV regression in column (4), and the method of one order difference is proposed before the IV regression in column (5). Further, we included

the underidentification test in the 2SLS model. The LM statistics in column (4) are 10.995, and the *p*-value is 0.0009. The LM statistics in column (5) are 5.503, and the *p*-value is 0.0199. The results of the underidentification test show that the instrument is correlated with the endogenous regressor. Column (6) is the result of the GMM dynamic panel regression. As the independent variable "rcli" is correlated with the lagged dependent variable, we replaced "rcli" with "dcli" in dynamic panel regression. The optimal lag order of the dynamic panel regression is 2 according to the AR test, and the GMM method was used to solve the endogeneity problem caused by the lagged dependent variable. Further, as there were 34 instruments, we included the overidentification tests. The *p*-value of the overidentification tests is 0.1521, which means that all instrumental variables are valid. All results in Table 8 show that implicit contingent liabilities have a negative effect on economic growth, so the results of the previous studies are robust.

**Table 8.** Robustness test.

| | **Dgdp** | | **rcli** | **dgdp** | | | | **dgdp** |
|---|---|---|---|---|---|---|---|---|
| | **(1)** | **(2)** | **(3)** | **(4)** | **(5)** | | | **(6)** |
| | Replace rcli by dcli | Replace rcli by rcli_iv | IV Regression (2SLS-1) | IV Regression (2SLS-2-FE) | IV Regression (2SLS-2-FD) | | | Dynamic Panel Regression (GMM) |
| - | - | - | - | - | - | L.dgdp | | −0.195 *** (0.0266) |
| - | - | - | - | - | - | L2.dgdp | | −0.290 *** (0.0194) |
| rcli | - | - | - | −0.231 *** (0.0750) | −0.242 ** (0.111) | rcli | | - |
| dcli | −0.0249 *** (0.00459) | - | - | - | - | dcli | | −0.0137 *** (0.00342) |
| rcli_iv | - | −0.222 *** (0.0345) | - | - | | rcli_iv | | - |
| gov | 0.406 *** (0.123) | 0.409 *** (0.121) | 0.797 *** (1.020) | 0.834 (0.279) | (0.562) | gov | | 0.141 * (0.0856) |
| dfdi | −0.00747 (0.00561) | −0.00529 (0.00549) | −0.0522 (0.0463) | −0.0173 (0.0122) | −0.0640 ** (0.0318) | dfdi | | −0.00480 (0.00325) |
| edu | −5.126 *** (1.527) | −4.957 *** (1.493) | −17.96 (12.58) | −9.099 *** (3.525) | −15.77 * (9.284) | edu | | −2.651 (2.007) |
| inv | −0.0299 (0.0247) | −0.0107 (0.0245) | 0.00436 (0.207) | −0.00969 (0.0515) | −0.0924 (0.109) | inv | | −0.0993 * (0.0535) |
| city | 0.636 *** (0.196) | 0.613 *** (0.191) | 2.996 * (1.609) | 1.304 *** (0.487) | 1.525 (1.200) | city | | 0.258 ** (0.104) |
| corruption | −0.557*** (0.0661) | −0.518*** (0.0650) | 0.167 (0.548) | −0.479*** (0.138) | −0.265 (0.275) | corruption | | −0.363 *** (0.0363) |
| lnpo | −0.00578 (0.0167) | −0.0327* (0.0174) | −0.135 (0.147) | −0.0640 (0.0414) | −0.0294 (0.129) | lnpo | | −0.0723 *** (0.0130) |
| policy | 0.00944 (0.0108) | 0.0102 (0.0106) | −0.0391 (0.0896) | 0.00116 (0.0225) | −0.0161 (0.0288) | policy | | −0.0358 *** (0.00325) |
| crisis | −0.0167** (0.00848) | −0.0106 (0.00844) | 0.0154 (0.0711) | −0.00700 (0.0180) | −0.0231 (0.0246) | crisis | | −0.0938 *** (0.00593) |
| Constant | −0.363** (0.172) | −0.106 (0.176) | −0.367 (1.480) | −0.190 (0.362) | −0.00469 (0.0266) | Constant | | 0.678 *** (0.126) |
| LM statistics | - | - | - | 10.995 (0.0009) | 5.503 (0.0190) | - | | - |
| Observations | 341 | 341 | 341 | 341 | 310 | Observations | | 248 |
| R-squared | 0.647 | 0.659 | 0.084 | - | - | R-squared | | - |
| Number of id | 31 | 31 | 31 | 31 | 31 | Number of id | | 31 |

Notes: 1. The number in parentheses is the cluster robust standard error; 2. *** indicates that the statistic is significant at the 1% level, ** indicates that the statistic is significant at the 5% level and * indicates that the statistic is significant at the 10% level. 3. R-squared has low reference value in the IV regression and dynamic panel regression, so it is generally not reported. 4. LM statistics is the results of underidentification test.

Table 9a,b test the robustness of the methods on how to balance leverage and economic growth. We replaced the variable of the efficiency of total capital with the variable of the efficiency of total debts, and the results of the robustness test are similar to that of previous studies. Table 9a,b show that transferring leverage and improving the efficiency of enterprises are both effective in promoting economic growth.

**Table 9.** Robustness test 2.

| | **(a) Robustness Test 2** | | | | |
|---|---|---|---|---|---|
| | **ttd** | | | | |
| | **(1)** | **(2)** | **(3)** | **(4)** | **(5)** |
| dpri*dpri | | | | | −2.452 * (1.442) |
| d_rstate | −2.300 *** (0.357) | | | | |
| d_rpri | | 2.300 *** (0.357) | | | 3.598 *** (0.843) |
| statettd | | | 1.289 *** (0.0640) | | |
| prittd | | | | 0.251 *** (0.0160) | |
| gov | 0.885 (0.588) | 0.885 (0.588) | −0.539 (0.415) | 0.878 * (0.466) | 0.960 (0.588) |
| dfdi | 0.0133 (0.0268) | 0.0133 (0.0268) | 0.0226 (0.0186) | −0.00759 (0.0213) | 0.0167 (0.0268) |
| edu | −0.590 (7.328) | −0.590 (7.328) | 5.701 (5.030) | 6.320 (5.728) | 0.135 (7.318) |
| inv | −0.114 (0.118) | −0.114 (0.118) | 0.262 *** (0.0830) | −0.180 * (0.0937) | −0.139 (0.119) |
| city | 2.527 *** (0.921) | 2.527 *** (0.921) | 0.662 (0.643) | 1.029 (0.731) | 2.476 *** (0.919) |
| corruption | −1.245 *** (0.316) | −1.245 *** (0.316) | 0.217 (0.229) | −0.614 ** (0.252) | −1.225 *** (0.315) |
| lnpo | 0.145 * (0.0797) | 0.145 * (0.0797) | 0.121 ** (0.0549) | 0.0847 (0.0629) | 0.147 * (0.0794) |
| policy | −0.100 * (0.0517) | −0.100 * (0.0517) | 0.00278 (0.0364) | −0.0441 (0.0412) | −0.0972 * (0.0516) |
| crisis | −0.00553 (0.0403) | −0.00553 (0.0403) | 0.0407 (0.0282) | −0.0259 (0.0320) | −0.00958 (0.0403) |
| Constant | 0.0816 (0.909) | −2.218 *** (0.822) | −1.366 ** (0.574) | −0.945 (0.657) | −2.335 *** (0.822) |
| Observations | 341 | 341 | 341 | 341 | 341 |
| R-squared | 0.278 | 0.278 | 0.651 | 0.647 | 0.285 |
| Number of id | 31 | 31 | 31 | 31 | 31 |
| | **(b) Robustness Test 2 with Lag Variables** | | | | |
| | **ttd** | | | | |
| | | **(1)** | **(2)** | **(3)** | **(4)** |
| d_rstate | | −2.295 *** (0.384) | | | |

**Table 9.** *Cont.*

| | (b) Robustness Test 2 with Lag Variables | | | |
|---|---|---|---|---|
| | **ttd** | | | |
| | **(1)** | **(2)** | **(3)** | **(4)** |
| d_rpri | | 2.295 *** | | |
| | | (0.384) | | |
| statettd | | | 0.904 *** | |
| | | | (0.0680) | |
| prittd | | | | 0.256 *** |
| | | | | (0.0189) |
| gov | 0.859 | 0.859 | 0.157 | 1.934 *** |
| | (0.594) | (0.594) | (0.501) | (0.501) |
| dfdi | 0.0125 | 0.0125 | 0.0437 * | 0.000775 |
| | (0.0270) | (0.0270) | (0.0227) | (0.0225) |
| edu | 2.491 | 2.491 | −5.599 | −7.956 |
| | (7.334) | (7.334) | (6.194) | (6.181) |
| inv | −0.0651 | −0.0651 | −0.0162 | −0.0820 |
| | (0.119) | (0.119) | (0.0995) | (0.0989) |
| city | 1.358 | 1.358 | 3.805 *** | 3.682 *** |
| | (0.935) | (0.935) | (0.788) | (0.782) |
| corruption | −1.035 *** | −1.035 *** | −1.330 *** | −1.354 *** |
| | (0.319) | (0.319) | (0.268) | (0.266) |
| lnpo | 0.149 * | 0.149 * | 0.195 *** | 0.00772 |
| | (0.0804) | (0.0804) | (0.0664) | (0.0681) |
| policy | −0.135 ** | −0.135 ** | 0.0206 | −0.0127 |
| | (0.0524) | (0.0524) | (0.0448) | (0.0440) |
| crisis | −0.0341 | −0.0341 | −0.0321 | −0.000372 |
| | (0.0409) | (0.0409) | (0.0342) | (0.0339) |
| Constant | 0.687 | −1.608 * | −3.660 *** | −2.053 *** |
| | (0.979) | (0.840) | (0.701) | (0.691) |
| Observations | 341 | 341 | 341 | 341 |
| R-squared | 0.265 | 0.265 | 0.483 | 0.489 |
| Number of id | 31 | 31 | 31 | 31 |

Notes: 1. The number in parentheses is the cluster robust standard error; 2. *** indicates that the statistic is significant at the 1% level, ** indicates that the statistic is significant at the 5% level and * indicates that the statistic is significant at the 10% level.

## 7. Main Findings and Suggestions

Taking the data of 488 local SOEs from 2007 to 2020, we constructed a measurement index of local governments' implicit contingent liabilities incurred by local SOEs. Then, we studied the influence of these implicit contingent liabilities on the economic growth and its mechanism. Our findings are as follows: first, an increase in local SOEs' assets reduces economic fluctuations but increases implicit contingent liabilities of local governments and has negative effects on economic growth; second, implicit contingent liabilities affect economic growth differently, through problem loans and the total efficiency of capital, to how explicit liabilities do; third, transferring leverages from local SOEs to non-SOEs can improve the efficiency of total capital and mitigate the negative effects of implicit contingent liabilities on economic growth, which can help balance leverage and economic growth.

Based on above conclusions, we put forward the following policy recommendations for China to reduce the risk of government implicit contingent liabilities and other developing countries to improve their economic stability:

Firstly, the leverage will be transferred from local SOEs to non-SOEs to reduce the debt of local SOEs. Research has shown that there is a positive relation between the overall efficiency of capital and local economic growth, and the efficiency of capital of non-SOEs is higher than that of SOEs. Therefore, transferring leverage to non-SOEs will improve the overall efficiency of capital and promote economic growth [84]. However, there is an inverted U-shaped relationship between the proportion of non-SOEs and the overall efficiency of capital, so transferring leverage is helpful for economic growth within a limit.

Secondly, local governments should promote the reformation of SOEs and improve their efficiency, and thus improve the overall efficiency of capital [85–87]. The results showed that the efficiency of SOEs contributes more to the overall efficiency of capital. In order to mitigate the negative impact of local governments' implicit contingent liabilities on economic growth, it is necessary to promote the reformation of SOEs and improve their efficiency. How to improve the efficiency of SOEs has already been an important topic in academic circles and government departments. At present, the mixed ownership reform of SOEs is one of the ways [5,88,89].

Thirdly, China needs to reform and form a fair and competitive financial market, break the "credit discrimination" of banks and other financial institutions against non-SOEs and let markets determine where the leverage goes [90,91]. Credits will flow to departments with higher efficiency. Further, fair competition in financial markets also helps to promote SOEs to improve the efficiency in order to obtain financial support from financial institutions and thus increase economic growth.

Fourthly, there are various sources of implicit contingent liabilities of local governments [15]. Therefore, local governments also need to pay attention to these other sources and analyze how and when implicit contingent liabilities become explicit. Then, they can enhance their capacity to reduce the potential risks of implicit contingent liabilities.

For other countries, given China's experience in maintaining economic stability and growth during many rounds of economic crises, on the one hand, other developing countries can develop and revive some SOEs and take them as a useful instrument for macroeconomy in times of crisis to help the country ride out the crisis smoothly and stabilize the economic cycle fluctuations [1,21,92]. On the other hand, SOEs should also improve the efficiency to maintain the competitiveness in the fair competition market environment. SOEs should stabilize the economy and reduce the distortion of the market environment caused by themselves and other negative effects at the same time.

**Author Contributions:** Conceptualization, Y.D and M.G.; methodology, Y.D. and Y.H.; software, Y.D.; validation, Y.D., M.G. and Y.H.; formal analysis, Y.D.; investigation Y.D.; resources, Y.D.; data curation, Y.D.; writing—original draft preparation, Y.D.; writing—review and editing, Y.D., M.G. and Y.H.; visualization, M.G.; supervision, M.G.; project administration, M.G.; funding acquisition, M.G. All authors have read and agreed to the published version of the manuscript.

**Funding:** This research was funded by the National Social Science Foundation of China, China grant number17BJY170.

**Institutional Review Board Statement:** Not applicable.

**Informed Consent Statement:** Not applicable.

**Data Availability Statement:** The data sources in this paper are Wind database (https://www.wind.com.cn/ accessed on 24 February 2022) and the National Bureau of Statistics of China (http://www.stats.gov.cn/ accessed on 24 February 2022).

**Conflicts of Interest:** The authors declare no conflict of interest.

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
