# Peer review of "Leverage of Local State-Owned Enterprises, Implicit Contingent Liabilities of Government and Economic Growth"

_sustainability, doi:10.3390/su14063481_

Round 1

Reviewer 1 Report

Reviewer’s Report

Title of Manuscript: Leverage of local state-owned enterprises, implicit contingent 2 liabilities of government and economic growth

Manuscript Number: sustainability-1607099                              

Report Date:                     2/10/2022

The authors are to investigate the contingent liabilities’ formation and influence. The authors propose that (1) The expansion of asset scale of local SOEs can reduce the economic fluctuations, but increases the implicit contingent liabilities of local governments (Hypothesis 1); (2) The implicit contingent liabilities have a negative impact on economic growth (Hypothesis 2a) and influence economic growth through private enterprises’ investments and banks’ credits (Hypothesis 2b); (3) The transfer of leverage from SOEs to private enterprises and local governments can improve the overall efficiency of capital, which can relieve the negative effects of the implicit contingent liabilities on economic growth (Hypothesis 3a) and the transfer only plays a positive role within an appropriate scope (Hypothesis 3b).  Using a sample of Chinese local SOEs during the period of 2007 to 2018, they show that the expansion of local SOEs’ assets suppress economic fluctuations at the cost of increasing government’s implicit contingent debts, and will have a negative impact on economic growth.

The authors investigate an interesting topic.  However, to fulfill its purpose, there are more work needed. Here are some of my concerns.

  1. Contribution

The authors argue that they construct an index to measure the implicit contingent liabilities of the governments derived from local state-owned enterprises, and expand the research scope of implicit debts, analyze the influence path of local governments’ implicit contingent debts on economic growth, and explore the ways of reducing governments’ debts originated from local state-own ed enterprises.  These are what they do in the study, but not contributions. The authors need to highlight what make their study different from the prior studies in the literature. What the earlier studies have done? What do they fail to accomplish? What make the current study unique and deserve more attention from the researchers?

In particular, the author use Chinese data to investigate the link between local SOEs, local government and local economy.  Since the journal has an international audience, it is necessary to justify why the Chinese data are used and what unique settings in which the data could provide the advantage of addressing the issues other settings could not test. 

In addition, the authors do not mention the implication of the findings.  To whom those findings would be helpful? Could other countries learn from these findings? How about other emerging markets?

  1. Hypotheses

An hypothesis should be a falsifiable statement. However, some of the hypotheses in the paper is too vague to reject and thus not possible to prove wrong. For instance, the authors propose that the transfer only plays a positive role within an appropriate scope (Hypothesis 3b).  What is an appropriate scope? How to define it? What about a negative role? Under which condition? Do you mean an inverse U-shape? If so, please clearly argue and present it in the hypotheses development.

  1. Conceptual issues and measures

In the paper the authors define those firms without government ownership (non-SOE) as POE, privately owned enterprises.  This is not consistent with the literature, which defines privately owned enterprises as firms whose stock shares are not publicly traded on stock exchanges. Therefore, I suggest the authors use the term of non-SOE to avoid confusions.

A proper term is Dependent Variables instead of Explained Variables, vs. Independent variables instead of Explaining variables or Explanatory variables. Please correct the terms throughout the paper.

Please make sure the variable list in Table 2 is complete. For instance, L.policy and L.time in Table 4 are not defined.

Many concepts are not clearly defined and thus confusing to readers. For instance, Corruption in Table 2 is defined as “Corruption governance indicators in WGI index published by WB”. However, the whole paper does not mention what is WGI or WB nor how the data are collected. Another example is dcli, which in Table 2 is defined as “Take the method in Arslanalp and Liao’s paper as reference”.  You could not use a reference as a definition of a variable.

It is not clear how the stdgdp is calculated. Within the recent three year? Or the whole sample period? Then how you calculate the dgdp? Since you define it as annual change in gdp, you should have more observations in analyzing dgdp than stdgdp.

  1. Methodology and empirical results

Table 1 should list both the numbers of non-SOEs and SOE for each province, since the authors use non-SOE data to calculate their variables (see variable descriptions in Table 2). Also lease list the total number of both non-SOEs and SOEs. Based on my observation, the total number of province or SOE observations in Table 1 does not agree with those in other tables (341 observations). It also disagrees with the statement in conclusion which mentions 490 observations. It seems the authors delete many observations due to the data availability or other reasons. 

Please add a detailed table on the sample selection process including the number of provinces, the number of SOEs and non-SOE firms (since you also use them to calculate your major variables) and the number of observations. Please add one table of sample distributions by year.

Table 3 should include median of variables. Variance should be changed to standard deviations.

Table 6 is confusing. The symbols are not clearly defined.  Is it a correlation table? If so, what variables are used?  I would expect a multivariate regression analysis here.

Table 6 shows a negative coefficient between economic growth and private enterprises’ investments and banks’ credits. The authors conclude that the contingent liabilities have negative impact on economic growth via private enterprises’ investments and banks’ credits (Hypothesis 2b). This conclusion is not persuasive. Why it is not the other way around: the economic growth affect private enterprises’ investments and banks’ credits? The authors need to use lag variables to figure out the causal relation.

Table 7 is confusing. Please use two panels to separately report the two parts.

The authors argue that transfer of leverage from SOEs to private enterprises and local governments can improve the overall efficiency of capital, which can relieve the negative effects of the implicit contingent liabilities on economic growth (Hypothesis 3a). A proper design would be a regression of economic growth on contingent liabilities, efficiency of capital and the interactive term between them. A negative coefficient on the interactive term would provide support to Hypothesis 3a.  However, I did not see such a design.  The same issue is for the testing of Hypothesis 3b. Combined with my comments above on Hypotheses, the authors either need to rewrite the hypotheses or redesign their tests.

The paper needs more discussions to align the findings with prior literature.  For instance, the literature indicates that sharp reduction of leverage may reduce the economic growth and even may lead to long-term recession (e.g., Roxburgh et al., 2010).  The authors argue that there could be a double edge of sword effect (although the hypotheses and results need to reorganized).  The authors need to provide detailed discussions on these issues in the revised version in highlighting these contributions to the theory and the empirical studies.

References

Roxburgh C, Lund S and Wimmer T,2010, “Debt and Deleveraging: The Global Credit Bubble and its Economic 699 Consequences”, McKinsey Global Institute, pp.1-94.

Reviewer 2 Report

In this manuscript, the authors construct the index of local governments’ implicit contingent liabilities based on the local SOEs, and study the implicit contingent liabilities’ formation and influence. The findings show that the expansion of local SOEs’ assets suppress economic fluctuations at the cost of increasing government’s implicit contingent debts, and will have a negative impact on economic growth. Different from the fiscal influence path of explicit debts, the implicit contingent debts restrain the local economic growth through market path. Deleveraging of local SOEs and improving the efficiency of local SOEs can effectively improve the overall efficiency of local funds and reduce the negative effect of local governments’ implicit contingent liabilities on economic growth. The followings are suggested for improvements of this manuscript.

  1. The authors should better explain the motivation of this manuscript. There are many research results on the problem of state-owned 29 enterprises (SOEs).
  2. There is a need for a literature review. A sufficient number of articles were included in the study. However, the studies used in the article need to be elaborated. For this, it is recommended to give detailed information about these studies. Thus, the reader will see the gap in the literature more clearly. Please, extend the discussion of the literature review included in your paper, highlighting the main contributions of each group of papers (it is not enough to include just a list of papers).
  3. The research progress of this theme needs to be more comprehensive, and some related papers need to be added: Dai, Z.F., Kang, J. Some new efficient mean-variance portfolio selection models, International Journal of Finance & Economics, 2021, 1-13. Dai, Z.F., Kang, J., Wen, F. Predicting stock returns: a risk measurement perspective. International Review of Financial Analysis, 2021, 74, 101676.
  4. The empirical findings section also needs to be improved. I must acknowledge that the section was developed to connect with past studies, which is impressive. However, this section still needs to be thoroughly developed. I recommend that the authors explain the reasons for similarities or differences between your findings and other studies' findings. The authors should also provide a deep analysis of the economic intuition behind the connectedness and the dependency rather than just statistical findings.
  5. The authors should better highlight the contribution of this manuscript. In particular, in the conclusion the author should give detail describe about it.

As a general conclusion, I recommend the authors to make a revision of their paper according to the above suggestions.

Reviewer 3 Report

  • There are several typos throughout the paper, e.g., line 52, 60, 104, 112 etc.
  • You should present in a timely manner and discuss in more detail previous studies on your topic
  • You should present firstly what specific abbreviations mean
  • How did you choose the values of loss given default to 40%, 50% or 60%? Give an explanation and present relevant literature
  • The text from line 133 to line 147 is redundant in that context and should be eliminated
  • Some citations are awkwardly presented, such as Mr Florio, 2013
  • The authors state that „Even though the number of companies in the sample is less than 1% of the number of all local SOEs, but their capital size is large. So, this is a representative sample.” They should develop this argument because having high capital level does not mean „large” but rather better capitalization. Rather, one would measure the size of the company using total assets. Thus, the authors should present relevant statistics to support their argument
  • You should describe all variables employed in the empirical analysis, starting with Eq. (6)
  • Is SOE fixed effects in the regression models? Or region fixed effects? You should clarify these aspects. Also, you do not include time fixed effects to account for common shocks, which would influence the results. You should add time fixed effects at least for robustness checks
  • It seems that you do not control for autocorrelation and heteroskedasticity in the residuals which may bias the results and reduce standard errors. Thus, you should apply specific tests to conform the rejection of autocorrelation and heteroskedasticity in the residuals, or use robust standard errors

Reviewer 4 Report

In the Abstract it is necessary to add information about the purpose of the study and the number of samples as well as sampling techniques.
References on the background should use the latest references, namely the last 5 years or a maximum of the last 10 years. Many manuscripts still use unupdated references (unless Grand Theory is allowed).
Not all paragraphs are supported by references, it is better to support them with updated references so that the arguments are strong.
In the Theoretical framework section, it is better to explain the theories (Grand Theory) that are used or underlie and can be explained before the explanation of the development of the hypothesis.
The time interval we studied is from 2007 to 2018, preferably up to 2020.
It is necessary to explain how observations obtained 341 results and Taking 490 local SOEs during the period.
Table 2. Variable descriptions should be completed and added a column for the source/reference.

Round 2

Reviewer 1 Report

I notice that the authors have made great effort in addressing the reviewers' comments. I would like the authors to make the following minor revisions:

Regarding the responses to Comments 1, 2 and 3, please cite prior studies to support your arguments. It is not proper to generally mention prior literature or assert a statement without a reference.

Regarding the response to Comment 8, please fully spell WGI and WB when you first mention them in the paper and then you could use the abbreviations.

Regarding Table 6 in Comment 13, there are still some issues. First, please remove Chinese Characters from the table. Second, please add the explanations of the symbols in a footnote below the table, including the approach you used. Third, please change "correlated" into "related", as the table is a multivariate regression not a correlation matrix. Correlation is only used in a univariate setting without controlling for other variables.

Regarding Comment 14, the argument is not persuasive. You could not define the link as time-limited effect without empirical evidence. Please add this as the limitation if you could not address the issue.

Reviewer 2 Report

The author has made the required corrections to the first revision. I recommend its publication.

Author Response

Thank you for your comments.

Reviewer 3 Report

  • In Eq. (6)-(12) coefficient for control variables should be specified as a vector. At this moment, you only have one coefficient for all control variables
  • Usually, one would put the table with descriptive statistics after the presentation of the empirical model, not before
  • What is epsiloni in the models described by Eq. (6)-(12)? SOE FE or region FE? You should clarify this aspect
  • Bellow each tabe, add specific goodness-of-fit tests, such as R-squared, F-statistics etc. Also, it’s not clear whether you cluster standard errors by SOE of by region
  • When you include interactions in you model, you should add full constitutive terms. This biases the results, as the interaction term could absorb the effect. See Brambor et al. (2006) for a more detailed explanation
  • In Table 6, last line, you have foreign language characters that are unintelligible. Pay more attention to the formatting
  • How did you choose rcli_iv as instrument? More specifically, what does it measure? Add a brief discussion on this issue as it is very difficult to find a valid instrument. A valid instrument should be correlated with your instrumented variable, but uncorrelated with the error term. These aspects are not discussed in the paper. Also, you should present overidentification and underidentification tests that formally test the validity of your instrument. Moreover, what model do you apply in your IV regression?
  • You should note that, by definition, in the dynamic model, the lagged dependent vriable is correlated with the error term. Thus, FE models are not adequate against this backdrop. Rather, you should apply GMM-type models when using dynamic models

References

Brambor, T., Clark, W.R., Golder, M., 2006. Understanding interaction models: improving empirical analysis. Political Analysis 14, 63-82

Reviewer 4 Report

Dear Author,

Some suggestions have been made. However there are still references that have not been added and updated, please pay special attention to this revision.

Introduction is still not strong, previous research support and references are not updated, make sure each paragraph has strong reference support. Still using old references and need to be updated, like:

  1. (Stern and Feldman,2004)
  2. HANA (1999)
  3. Cebotari et al. (2009)
  4. (Hana, 1999; Liu Shangxi, 2003)
  5. (Liu Yuanchun, 2001)
  6. (Megginson and Netter, 2001
  7. Djankov and Murrel, 2002
  8. Tan Jinsong and Zheng Guojian, 2004
  9. Song Li-gang, Yao Yang, 2005
  10. Liu Ruiming and Shi Lei, 2010
  11. (Dotsey, 1994
  12. Gale and Orszag, 2003
  13. (Bruce and Laski,1998)
  14. Liu Yuanchun, 2001
  15. Sun and Tong, 2003
  16. Zhang et al., 2001
  17. Liu Shangxi et al., 2003
  18. Fisher, 1933
  19. Kornai, 1986
  20. Shleifer and Vishny,1994
  21. Dewatripont and Maskin, 1995
  22. Aschauer, 1989)

and others can be checked, especially those that are more than the last 10 years. It is expected that the reference is the latest in the last 5 years.

In the discussion, references need to be strengthened and in the manuscript there are still minimal references so that the discussion is not strong.

Round 3

Reviewer 3 Report

  • In the 2SLS model, you should include the underidentification test, i.e., testing formally that the excluded instrument is correlated with the endogenous regressor

  • In the GMM specification, is not discussed how you address the endogeneity of the lagged dependent variable, and thus the number of instruments employed. Also, you should include AR(1) and AR(2) and overidentification tests here
